

# Predicting the results of evaluation procedures of academics

Francesco Poggi[1], Paolo Ciancarini[1,2], Aldo Gangemi[3], Andrea Giovanni Nuzzolese[4], Silvio Peroni[3] and Valentina Presutti[4]

[1] Department of Computer Science and Engineering (DISI), University of Bologna, Bologna, Italy
[2] Institute of Data Science and Artificial Intelligence, Innopolis University, Innopolis, Russia
[3] Department of Classical Philology and Italian Studies, University of Bologna, Bologna, Italy
[4] STLab, Institute of Cognitive Science and Technologies, National Research Council, Roma, Italy

## ABSTRACT

**Background**. The 2010 reform of the Italian university system introduced the National Scientific Habilitation (ASN) as a requirement for applying to permanent professor positions. Since the CVs of the 59,149 candidates and the results of their assessments have been made publicly available, the ASN constitutes an opportunity to perform analyses about a nation-wide evaluation process.

**Objective**. The main goals of this paper are: (i) predicting the ASN results using the information contained in the candidates' CVs; (ii) identifying a small set of quantitative indicators that can be used to perform accurate predictions.

**Approach**. Semantic technologies are used to extract, systematize and enrich the information contained in the applicants' CVs, and machine learning methods are used to predict the ASN results and to identify a subset of relevant predictors.

**Results**. For predicting the success in the role of associate professor, our best models using all and the top 15 predictors make accurate predictions (F-measure values higher than 0.6) in 88% and 88.6% of the cases, respectively. Similar results have been achieved for the role of full professor.

**Evaluation**. The proposed approach outperforms the other models developed to predict the results of researchers' evaluation procedures.

**Conclusions**. Such results allow the development of an automated system for supporting both candidates and committees in the future ASN sessions and other scholars' evaluation procedures.

## INTRODUCTION

Quantitative indicators have been extensively used for evaluating scientific performances of a given research body. International institutions, national authorities, research and funding bodies have an increasing interest on indicators, mainly based on bibliometric data, which can be used to algorithmically assess the performance of their institutions. SCImago (https://www.scimagojr.com/) (for journals), the Performance Ranking of Scientific Papers for World Universities (http://nturanking.lis.ntu.edu.tw/) and the Academic Ranking of

Corresponding author
Francesco Poggi, fpoggi@cs.unibo.it, francesco.poggi5@unibo.it

World Universities (http://www.shanghairanking.com/) (for universities) are popular examples of rankings that use bibliometric indicators to rate scientific performances.

Peer review is still the Holy Grail for research evaluation, but the pressure for more frequent and extensive assessments of the performance of researchers, research groups and institutions makes bibliometry attractive. Currently, several countries use a combination of peer review and bibliometric indicators to allocate funding and evaluate the performance of higher education institutions. Examples of this mixed strategy are the Excellence in Research for Australia (ERA) and the Valutazione della Qualità della Ricerca (VQR) in Italy. The British Research Excellence Framework (REF), successor of the Research Assessment Exercise (RAE), is another example, in which experts can make use of citation data as an additional input of their reviews. In many countries, bibliometric indicators are one of the factors that can be used for assessing individuals or institutions to allocate funding at a national level. For instance, in Germany the impact factor of the publications is used in performance-based funding systems, in Finland, the reallocation system uses the number of publications as one of the considered measures, in Norway, a two-level bibliometric indicator is used for similar purposes, etc. (*Vieira, Cabral & Gomes, 2014a*).

The growing importance of quantitative indicators may be mainly explained by their advantages compared to peer review processes: objectivity, low time and implementation costs, possibility of quick and cheap updates, ability to cover a large number of individuals, etc. However, in many cases peer review is still the only method available in practice, and is hence intensively used in many situations. We know that bibliometric indicators are more accepted in the assessment of large research bodies, but they are still used frequently for individuals. It is, therefore, very important to benchmark bibliometric indicators against traditional peer assessments in real situations.

Some studies have been carried out in recent years with the main goal of finding a relation between the two methods at several levels. At national level, the relation between bibliometric indicators and the results of the Research Assessment Exercise (RAE) in Britain (*Norris & Oppenheim, 2003*; *Taylor, 2011*) or the Italian Triennial Assessment Exercise (VTR) (*Abramo, D'Angelo & Caprasecca, 2009*; *Franceschet & Costantini, 2011*) have been investigated. Other studies focused on the assessments of departments (*Aksnes, 2003*) and research groups (*Van Raan, 2006*). Just a few works have been made at the individual level (*Nederhof & Van Raan, 1987*; *Bornmann & Daniel, 2006*; *Bornmann, Wallon & Ledin, 2008*), while many analyzed the correlation between indicators and research performances (*Leydesdorff, 2009*; *Franceschet, 2009*). Recent works analyzed the correlation between traditional bibliometric indicators and altmetrics by also taking into account quality assessment procedures performed by peers (*Nuzzolese et al., 2019*; *Wouters et al., 2015*; *Bornmann & Haunschild, 2018*). All these works share the general finding that a positive and significant correlation exists between peer review and bibliometric indicators, and suggest that indicators can be useful tools to support peer reviews.

In this work, we investigate the relation between quantitative indicators and peer review processes from a different perspective. The focus of the study is to analyze if and to what extent **quantitative indicators can be used to predict the results of peer reviews**. This problem is interesting for many different reasons. First of all, since a high number of

factors are involved in peer review processes (e.g., cultural, social, contextual, scientific, etc.), the feasibility of reproducing such a complex human process through computational and automatic methods is a relevant topic per se. Moreover, the possibility of predicting human assessments has many practical applications. Having an idea of the results of an evaluation procedure may be very useful for candidates (e.g., to understand if they are competitive for a given position, to decide to apply or not, etc.). Also, evaluators can benefit from such information (e.g., for supporting a first screening of the candidates, for spotting possible errors to investigate, etc.). In other words, the final goal of our work is not substituting peer committees with automatic agents, but **providing tools for supporting both candidates and evaluators in their tasks**.

[1] The acronym ASN stands for *Abilitazione Scientifica Nazionale*. For the rest of the paper, all acronyms (e.g., ASN, MIUR, ANVUR, etc.) are based on the original Italian names, since they are well established in the Italian scientific community. The English translations are also provided for the benefit of the international readers.

This study analyzes the Italian National Scientific Habilitation (ASN[1]), a nation-wide research assessment procedure involving a large number of applicants from all academic areas. The ASN is one of the main novelties in the national university system introduced by Law 240/2010 (*Law, 2011*), and it is similar to other habilitation procedures already in place in other countries (e.g., France and Germany) in that it is a prerequisite for becoming a university professor. The ASN is meant to attest that an individual has reached the scientific maturity required for applying for a specific role (associate or full professor) in a given scientific discipline; however, the qualification does not guarantee that a professorship position will eventually be granted. The assessments of the candidates of each discipline are performed by committees composed of four full professors from Italian universities and one professor from a foreign research institution. The evaluation is performed considering the CVs submitted by the applicants and three quantitative indicators computed for each candidate.

The first session of the ASN started on November 2012 and received 59,149 applications spanning 184 Recruitment Fields (RFs), which correspond to scientific fields of study in which Scientific Areas (SAs) are organized. The curricula of all applicants, the values of their bibliometric indicators and the final reports of examination committees have been made publicly available. This work focuses on the analysis of applicants' curricula. For this purpose, we processed this vast text corpus, extracted the contained information and used it to populate a Knowledge Graph by exploiting semantic technologies. This Knowledge Graph contains a collection of relevant data for each applicant and it has then been used to perform different kinds of analyses at the level of category of discipline (i.e., *bibliometric* and *non-bibliometric*), SA, and RF.

An approach based on machine learning techniques has been used to answer the following research questions:

- *RQ1:* Is it possible to predict the results of the ASN using only the information contained in the candidates' CVs?
- *RQ2:* Is it possible to identify a small set of predictors that can be used to predict the ASN results?

The rest of the work is organized as follows. 'Related Work' presents an overview of the related work. 'Methods and Material' provides necessary background information about the ASN, gives an overview of the ASN dataset, and describes the algorithms used in this

**Table 1  Comparison of the related work with our study.** Missing data are labeled with "n.a.". PoC stands for "Prediction of Citations", AoH for "Analysis of H-index for peer judgements", and PoPJ for "Prediction of Peer Judgements".

| Work | Papers | Authors | Discipline | Predictors | Task | Method |
|------|--------|---------|------------|------------|------|--------|
| *Ibáñez et al.* (JASIST, *2016*) | n.a. | 280 | Computer Science | 12 | PoC | Gaussian Bayesian networks |
| *Danell* (JASIST, *2011*) | 6,030 | 8,149 | Neuroscience and Physics | 2 | PoC | Quantile regression |
| *Fu & Aliferis* (Scientometrics, *2010*) | 3,788 | n.a. | Medicine | 12 (+ textual features) | PoC | Support vector machines |
| *Lindahl* (J. of Informetrics, *2018*) | n.a. | 406 | Mathematics | 4 | PoC | Logistic regression |
| *Bornmann & Daniel* (J. of Informetrics, *2007*) | n.a. | 414 | Biomedicine | 1 | AoH | Correlation analysis |
| *Van Raan* (Scientometrics, *2006*) | n.a. | 700 | Chemistry | 1 | AoH | Correlation and error analysis |
| *Cronin & Meho* (JASIST, *2006*) | n.a. | 31 | Information Science | 1 | AoH | Correlation analysis |
| *Vieira, Cabral & Gomes* (JASIST, *2014a*) | 7,654 | 174 | Hard sciences | 3 (based on 12 bibl. indices) | PoPJ | Rank ordered logistic regression |
| *Jensen, Rouquier & Croissant* (Scientometrics, *2009*) | n.a. | 3,659 | All | 8 | PoPJ | Binomial regression |
| *Tregellas et al.* (PeerJ, *2018*) | n.a. | 363 | Biomedicine | 10 (3 for the best model) | PoPJ | Logistic regression, Support vector machines |
| This work | 1,910,873 | 59,149 | All | 326 | PoPJ | Support vector machines (CFS for feature selection) |

work. In 'Results' we describe the results of the analyses performed to answer the two aforementioned research questions, and we evaluate our work by comparing the predictive power of our approach with others at the state of the art. Finally, in the last two sections we discuss the results and draw some conclusions.

## RELATED WORK

Quantitative indicators have been extensively used for evaluating the scientific performance of a given research body. Many recent studies have focused on the predictive power of such indicators for different purposes. These works can be divided into two main groups: those that use bibliometric indicators to predict other indicators and those that use bibliometric indicators to predict the results of evaluation procedures performed through a peer review process or a mixed strategy (i.e., a combination of peer review and bibliometric indicators). We discuss the main recent works on this topic. To facilitate the readers, Table 1 summarizes the main information about them and our study.

A first challenge concerns the problem of identifying a subset of bibliometric indicators for predicting other bibliometric indices. *Ibáñez et al.* (*2016*) introduced an approach based on Gaussian Bayesian networks to identify the best subset of predictive variables. The

approach has been tested on the data of 280 Spanish full professors of Computer Science using 12 bibliometric indicators. The main drawback of the work is that no evaluation is presented: only a test on a small sample composed of three cases is discussed in the paper. Other works focused on the prediction of papers citations. *Danell* (*2011*) used previous publication volume and citation rate of authors to predict the impact of their articles. The aim of this work is to investigate whether evaluations systems based on researchers' track records actually reward excellence. The study focused on two disciplines (i.e., episodic memory research and Bose–Einstein condensate) and developed a quantile regression model based on previous publication volume and citation rate to predict authors' relative citation rate. Another work (*Fu & Aliferis, 2010*) faces the problem of predicting the number of citations that a paper will receive using only the information available at publication time. The used model is based on support vector machines, and has been tested on a mixture of bibliometric features and content-based features extracted from 3788 biomedical articles. A recent work (*Lindahl, 2018*) investigates the ability of four indices to predict whether an author will attain excellence—operationalized by the indicator defined in (*Bornmann, 2013*)—in the following four years. The developed model is based on logistic regression and has been tested on a dataset composed of the track records of 406 mathematicians.

Only a few works focused on the problem of using bibliometric indicators to predict the results of evaluation procedures performed through peer-review processes. *Vieira, Cabral & Gomes* (*2014a*) compare three models for predicting the success of applicants to academic positions. The test dataset is composed of the track records of 174 candidates to 27 selection processes for associate and full professor in hard sciences that took place in Portugal between 2007 and 2011. The areas of Chemistry, Physics, Biology, Mathematics, Mechanics, Geology, and Computer Science were considered. In all cases, candidates have been assessed by a panel of peers, producing a ranking of the applicants. Starting from 12 bibliometric indicators (i.e., number of documents, percentage of cited, highly cited and citing documents, average number of authors, $h_{nf}$-index, NIR, SNIP, SJR, percentage of international collaborations, normalized impact and the number of Scimago's Q1 journals) a few composite indices have been derived through a factor analysis. Following a discrete choice model, three predictive models based on Rank Ordered Logistic Regression (ROLR) have been defined. The best model is able to predict the applicants placed in the first position by peers in 56% of the cases. By considering the problem of predicting the relative position of two candidates (i.e., who will be ranked in the higher position), the best model is able to predict 76% of the orderings. In another work (*Vieira, Cabral & Gomes, 2014b*), the performances of these models have been compared with a random model, observing that in 78% of the cases the applicant placed in first position by peers has a probability of being placed first that is better than chance. The authors conclude that the predictions provided by the models are satisfactory, and suggest that they can be used as an auxiliary instrument to support peer judgments.

Another work tested the predictive power of eight indicators for predicting scientists promotions (*Jensen, Rouquier & Croissant, 2009*). The dataset used in the study is composed of the track records of 3,659 CNRS researchers from all disciplines that have filled the CNRS report between 2005 and 2008, whose data has been obtained by querying the Web of Science

database. In the same timespan, the promotions of about 600 CNRS researchers at all the five CNRS levels have been considered. A binomial regression model (logit) has been used to assess the overall relevance of eight quantitative indicators (h-index, normalized h-index, number of publications and citations, mean citations per paper, h-index per paper, age, gender) and to study their dependence. The results showed that the h-index is the best index for predicting the promotions, followed by the number of publications. Differences exist between disciplines: in Engineering, for instance, the number of publications is the best predictor. A logit model based on the best overall predictor (i.e., h-index) has been tested for each subdiscipline, leading to correct predictions in 48% of the cases. The authors conclude that bibliometric indicators do much better than randomness, which would achieve 30% of guessed promotions.

A recent study (*Tregellas et al., 2018*) focused on the problem of predicting career outcomes of academics using the information in their publication records. The objective of the work is to identify the main factors that may predict the success of young researchers in obtaining tenure-track faculty research positions. The dataset used in this study is composed of the track records of 363 PhD graduates from biomedical sciences programs at the University of Colorado from 2000 to 2015. The ratio of faculty/non-faculty members (i.e., individuals employed/not employed in faculty positions) is 12%. For each PhD graduate, 10 indicators has been computed (i.e., sex, date of graduation, number of first-author and non-first-author publications , average impact factor of first-author and non-first-author publications, highest impact factor of first-author and non-first-author publications, weighted first-author and non-first-author publication count). Logistic regression models and support vector machines has been used to investigate and compare the ability of the aforementioned indicators to predict career outcomes. The best prediction has been performed by the logistic regression model using three predictors (i.e., sex, date of graduation, and weighted first-author publication count), showing 73% accuracy . A similar result (i.e., 71% accuracy) has been obtained by the best model based on support vector machines using the same predictors. The results suggest that, while sex and months since graduation also predict career outcomes, a strong predoctoral first-author publication record may increase the likelihood of obtaining an academic faculty research position. The analysis of the results also showed for all models high negative predictive values (i.e., high accuracy in predicting those who will not obtain a faculty position), while low positive predictive values. This suggests that first-author publications are necessary but not sufficient for obtaining a faculty position. The main limitation of the study concerns the dataset size, since it was conducted on a small set of individuals at only one institution, focusing on a single discipline. The authors observe that it is then necessary to determine how generalizable the current findings are. Finally, the fact that all the best models are less than 75% accurate suggests that variables other than those considered here are also likely to be important factors in predicting future faculty status.

Other empirical studies focused on a single indicator (i.e., the h-index) to assess how it correlates with peer judgements. These works have the main limitation of being carried out on small samples for technical reasons (i.e., the difficulty of obtaining large sets of robust bibliometric data). In practice, they were generally limited to a single discipline: *Bornmann*

*& Daniel* (*2007*) studied 414 applications to long-term fellowships in biomedicine, *Van Raan* (*2006*) analyzed the evaluation of about 700 researchers in chemistry, *Cronin & Meho* (*2006*) studied 31 influential information scientists from the US.

To the best of our knowledge, no other work analyzed the predictive power of quantitative indicators for predicting the results of peer judgments of researchers.

# METHODS AND MATERIAL

This section provides necessary background information about the ASN and describes the ASN dataset, the techniques used to analyze this text corpus, and the ontology developed for storing data in a semantic format. A description of the classification and feature selection algorithms used in the analyses presented in "Results" concludes the section.

## Data from the Italian Scientific Habilitation
### Background

The Italian Law 240/2010 (*Law, 2011*) introduced substantial changes in the national university system. Before 2010, in the Italian universities there were three types of tenured positions: assistant professor, associate professor and full professor. The reform suppressed the position of assistant professor and replaced it with two types of fixed term positions called type A and type B researcher. Type A positions last for three years and can be extended for other two years. Type B positions last for three years and have been conceived as a step for becoming tenured associate professor, since at the time of recruitment universities must allocate resources and funding for the promotion. Each academic is bound to a specific Recruitment Field (RF), which corresponds to a scientific field of study. RFs are organized in groups, which are in turn sorted into 14 Scientific Areas (SAs). In this taxonomy defined by Decree 159 (*Ministerial Decree 159, 2012*), each of the 184 RFs is identified by an alphanumeric code in the form AA/GF, where AA is the ID of the SA (in the range 01-14), G is a single letter identifying the group of RFs, and F is a digit denoting the RF. For example, the code of the RF "Neurology" is 06/D5, which belongs to the group "Specialized Clinical Medicine" (06/D), which is part of the SA "Medicine" (06). The 14 SAs are listed in Table 2, and the 184 RFs are listed in (*Poggi et al., 2018b*).

Under the new law, only people that attained the National Scientific Habilitation (ASN) can apply for tenured positions in the Italian university system. It is important to note that an habilitation does not guarantee any position by itself. The ASN has indeed been conceived to attest the scientific maturity of researchers and is a requirement for accessing to a professorship in a given RF. Each university is responsible for creating new positions for a given RF and professional level provided that financial and administrative requirements are met, and handles the hiring process following local regulations and guidelines.

The first two sessions of the ASN took place in 2012 and 2013. Although the Law 240/2010 prescribes that the ASN must be held at least once a year, the next sessions took place in 2016 (1 session), 2017 (2 sessions) and 2018 (2 sessions). At the time of the writing of this article, the last session of the 2018 ASN was still in progress, and the dates of the next sessions have not yet been set. For each of the 184 RFs, the Ministry of University and Research (MIUR) appoints an examination committee for the evaluation of

**Table 2** **The 14 Italian scientific areas.** For each we report the numeric ID, a three-letter code, the name of the area and the number of RFs it contains.

| ID | Code | Area name | N. of recr. fields |
|----|------|-----------|--------------------|
| 01 | MCS | Mathematics and Computer Sciences | 7 |
| 02 | PHY | Physics | 6 |
| 03 | CHE | Chemistry | 8 |
| 04 | EAS | Earth Sciences | 4 |
| 05 | BIO | Biology | 13 |
| 06 | MED | Medical Sciences | 26 |
| 07 | AVM | Agricultural Sciences and Veterinary Medicine | 14 |
| 08 | CEA | Civil Engineering and Architecture | 12 |
| 09 | IIE | Industrial and Information Engineering | 20 |
| 10 | APL | Antiquities, Philology, Literary Studies, Art History | 19 |
| 11 | HPP | History, Philosophy, Pedagogy and Psychology | 17 |
| 12 | LAW | Law | 16 |
| 13 | ECS | Economics and Statistics | 15 |
| 14 | PSS | Political and Social Sciences | 7 |
|    | **Total** | | **184** |

the candidates. The committees are composed of five full professors who are responsible for the evaluation of the applicants for associate and full professor. Committee members are randomly selected from a list of eligible professors, for a total of 920 professors. Different committees have been appointed for 2012, 2013 and 2016-18 sessions, respectively. In order to apply to a session of the ASN, candidates have to submit a curriculum vitae with detailed information about their research activities. Although the ASN is bound to a specific RF and professional level, it is possible to apply in different RFs and roles. In 2012, for example, 136/260 (52.3%) applicants for full professor in the RF 09/H1 (Information Processing Systems) also applied to 01/B1 (Informatics). Those who fail to get an habilitation cannot apply again to the same RF and level in the next session. Once acquired, an habilitation lasts for six years.

The ASN introduced two types of parameters called *bibliometric* and *non-bibliometric* indicators, respectively. Bibliometric indicators apply to scientific disciplines for which reliable citation databases exist. The three bibliometric indicators are:

- Normalized number of journal papers
- Total number of citations received
- Normalized h-index.

Since citations and paper count increase over time, normalization based on the scientific age (the number of years since the first publication) is used to compute most of the indicators. The aforementioned indicators are used for all RFs belonging to the first nine SAs (01-09), with the exception of the RFs 08/C1, 08/D1, 08/E1, 08/E2, 08/F1 and the four RFs belonging to the group Psichology (11/E). These RFs are collectively denoted as *bibliometric disciplines*.

**Table 3  The number of applications for associate and full professor for each session of the ASN.**

| Session | Associate professor | Full professor | Total |
|---|---|---|---|
| 2012 | 41,088 | 18,061 | 59,149 |
| 2013 | 11,405 | 5,013 | 16,418 |
| 2016 | 13,119 | 7,211 | 20,330 |
| 2017a | 3,254 | 1,515 | 4,769 |
| 2017b | 2,501 | 1,322 | 3,823 |
| 2018a | 5,176 | 2,445 | 7,261 |
| **Total** | **76,543** | **35,567** | **112,110** |

Non-bibliometric indicators apply for the RFs for which MIUR assessed that citation databases are not "sufficiently complete", and hence bibliometric indices can not be reliably computed. The three non-bibliometric indicators are:

- Normalized number of published books
- Normalized number of book chapters and journal papers
- Normalized number of paper published on "top" journals.

These are used for all RFs belonging to the last five SAs (10–14) with the exceptions described above. These RFs are denoted as *non-bibliometric* disciplines. It is important to remark that this terminology (i.e., "bibliometric" and "non-bibliometric") is used in the official MIUR documents but it is not consistent with that used by the scientometric community. Non-bibliometric indicators, for instance, are indeed bibliometric being based on paper counts. Given that these terms became standard within the Italian research community, we will follow the MIUR "newspeak" according to the definitions above.

The values of the indicators for each candidate were computed by the National Agency for the Assessment of Universities and Research (ANVUR), a public agency established with the objective of assessing Italian academic research. Data from Scopus and Web of Science were used for this computation, and only publications in a time window of ten years before the ASN session were considered. The computed indicators and the candidates' CVs are the only information provided to the evaluation committees for their assessments. The sessions of the ASN have been analyzed by a quantitative point of view in *Marzolla (2015)*, *Peroni et al. (2019)* and *Di Iorio, Peroni & Poggi (2019)*.

### ASN data

The number of applications submitted to the six sessions of the ASN are reported in Table 3. We focused on the 2012 session of the ASN because: (i) it is a representative sample of the whole population asking for habilitation (this session was the first and received the more than half of the overall submissions across all years of ASN); (ii) since in 2016 different people were appointed in the committees, in this way we exclude biases and other problems introduced by changes in the evaluation committees.

Overall, the 2012 session of the ASN received 59,149 applications spanning 184 RFs. For each application, we collected three different documents: the CV, the official document

with the values of the three quantitative indicators described in the previous section and the final reports of the examination committee. These documents are in PDF, and have been made publicly available on the ANVUR site for a short period of time. Some basic information and statistics about the 2012 ASN session are summarized in (*Poggi et al., 2018b*).

Since ANVUR did not provide a template for the habilitation, the CVs are very heterogeneous, varying in terms of formatting, internal structure and organization. This heterogeneity and the massive amount of information contained in the 59,149 PDFs are two of the main challenges faced in this work. In order to manage this problem, we developed an ontology which provides an uniform representation of the information and a reference conceptual model. It is the basis of both the data processing and subsequent analyses, as described in the following sections.

## Ontology description

The objective of the Academic Career (AC) ontology is to model the academic career of scholars. AC is an OWL2 (*W3C, 2012*) ontology composed of fifteen modules, each of which is responsible for representing a particular aspect of the scientific career of a scholar. The first two modules of the AC ontology concern personal information and publications. The next modules pertain to ten categories suggested by ANVUR:

1. Participation to scientific events with specific roles (eg. speaker, organizer, attendee, etc.)
2. Involvement and roles in research groups (management, membership, etc.)
3. Responsibility for studies and researches granted by qualified institutions
4. Scientific responsibility for research projects
5. Direction or participation to editorial committees
6. Academic and professional roles
7. Teaching or research assignments (fellowships) at qualified institutes
8. Prizes and awards for scientific activities
9. Results of technological transfer activities (e.g., spin-offs, patents, etc.)
10. Other working and research experiences

The last three modules concern scholars' education, scientific qualifications, and personal skills and expertise.

## Data processing

The processing of a vast set of documents such as the corpus of the ASN curricula is not a trivial task. The main issue to face in this process is the management and harmonization of its heterogeneity in terms of kinds of information, structures (e.g., tables, lists, free text), styles, languages, just to cite a few. Nonetheless, the automatic extraction of information from CVs and its systematization in a machine processable format is a fundamental step for this work, since all the analyses described in "Results" are based on these data. For this purpose, we developed PDF to Academic Career Ontology (PACO), a software tool that is able to process the researchers' CVs, extract the most relevant information, and produce a Knowledge Graph that conforms to the AC ontology. The processing performed

**Figure 1** **An overview of the PACO toolchain composed of four sub-modules (circles). Artifacts (i.e., inputs/outputs of the sub-modules) are depicted as rectangles.**

by PACO is composed of four consecutive steps, that correspond to the software modules constituting PACO's architecture, as shown in Fig. 1. The processing of an applicant's CV can be summarized as follows:

- **HTML conversion:** The *PDF2HTML Converter* takes as input a PDF and produces as output an HTML version of the CV composed of inline elements and presentational elements. The structure of the document is not reconstructed in this phase. In particular, the containment relations between elements (e.g., cells in a table, items in a list, etc.) are missing. For instance, a table is converted into a series of rectangles with borders (the cells) followed by a series of inline elements (the text). All the elements are at the same level in the output document hierarchy, and no explicit relation between them is maintained.

- **Structure re-construction:** the *Structure Builder* uses the presentational information computed in the previous phase to infer the structure of the document. Different strategies have been developed to recognize meaningful patterns in the presentation and reconstruct the document hierarchy. For example, a mark positioned near an inline element containing text is interpreted as a list item, a sequence of consecutive list items is interpreted as a list. The output is an XML document, in which the original textual content is organized in meaningful structural elements.

- **Semantic analysis:** the objective of the *Semantic Analyzer* is to annotate the output of the previous phase with information about its content. For example, it has to infer if a list is a list of publications, awards, projects, etc. A series of analyses is performed for each element, from simple ones (e.g., to test if an element contains a name, surname, birth date, etc.) implemented through basic techniques such as the use of heuristics or pattern matching, to more complex ones (e.g., to identify publications, roles, etc.) implemented using external tools and libraries. Another important technique is to leverage the homogeneity of structured elements (e.g., of all the items in a list or all the cells of a column) to infer meaningful information about their content, using the approach described in (*Poggi, Cigna & Nuzzolese, 2016*). The basic idea is that, for instance, if the majority of the elements of a list have been recognized as publications, it is then reasonable to conclude that also the others are publications. The output of this phase is an XML document annotated with the results of the semantic analysis.

- **Triplification:** the *Triplifier* is responsible for populating a Knowledge Graph with the information inferred in the previous phase. The marked XML document is the input of this stage, and the output is a Knowledge Graph that conforms to the AC ontology.

[2]http://cercauniversita.cineca.it/ is a MIUR service that provides information and statics about Italian professors, universities, degree programs, students, fundings, etc.

[3]TAking STock: external engagement by academics (TASTE) is an European project founded under the FP7 program that developed a database with data about the relation between universities and enterprises in Italy—see https://eventi.unibo.it/taste.

[4]Semantic Scout is a service that provides CNR scientific and administrative data in a semantic format—see http://stlab.istc.cnr.it/stlab/project/semantic-scout/.

The data extracted from the applicants' CVs by PACO have also been semantically enriched with information from the following external sources:

- Cercauniversita[2] : for information about the candidates' careers within the Italian university system;
- TASTE database[3] : for data about researchers' entrepreneurship and industrial activities from the TASTE database;
- Semantic Scout[4] : for information about researchers of the Italian National Council of Research (CNR).

The final outcome of this process is the Knowledge Graph from which we computed the predictors used in the analyses discussed in the following of this paper.

## Identification of the prediction algorithm

In order to implement a supervised learning approach, we needed to create a training set in which the ground truth is obtained from the final reports of the examination committees. The instances of our dataset correspond to the 59,149 applications submitted to the 2012 ASN. For each instance, we collected 326 predictors, 309 of which are numeric and 17 are nominal. The only source of data used to build our dataset is the Knowledge Graph containing the data extracted from the applicants' curricula and enriched with external information.

The predictors that have been computed belong to one of the following two categories:

- numeric and nominal values extracted from the CVs (e.g., the number of publications) or derived from the CVs using external sources (e.g., the number of journal papers has been computed using the publication list in the CVs and querying online databases like Scopus);
- quantitative values calculated using the values from the previous point. For example, we computed statistical indicators such as the variance of the number of journal papers for each applicant in the last N years.

The aforementioned 326 predictors and the habilitation class feature are our starting point to investigate the performances of different machine learning approaches. We decided not to explicitly split the dataset in training and test sets, and systematically rely on cross-fold validation instead. In particular, the data reported in this work are related to the 10-fold validation, but we have also performed a 3-fold one with very similar results. The following supervised machine learning algorithms have been tested:

- **NB:** Naïve Bayes (*John & Langley, 1995*)
- **KN:** K-nearest neighbours classifier (K chosen using cross validation) (*Aha, Kibler & Albert, 1991*)
- **C45:** C4.5 decision tree (unpruned) (*Quinlan, 2014*)
- **RandF:** Random Forest (*Breiman, 2001*)
- **SVM:** Support Vector Machine trained with sequential minimal optimization (*Keerthi et al., 2001*).

**Table 4** **Performance of the machine learning algorithms investigated for the classification of the applicants to the RF 11/E4 (level II).** For each algorithm we report Precision, Recall and F-Measure values.

|  | Precision | Recall | F-measure |
|---|---|---|---|
| NB | 0.856 | 0.850 | 0.853 |
| KN | 0.867 | 0.906 | 0.886 |
| C45 | 0.865 | 0.914 | 0.888 |
| RandF | 0.844 | 1.000 | 0.916 |
| SVM | 0.894 | 0.951 | 0.922 |

The rationale behind this choice is to have representatives for the main classification methods that have shown effectiveness in past research. All learners have been tuned using common best practices. SVM has been tested with various kernels (in order to account for complex non-linear separating hyperplanes). However, the best results were obtained with a relatively simple polynomial kernel. The parameters for the resulting model have been tuned using the grid method (*He & Garcia, 2009*). We tested the learners on different data samples obtaining similar results for both bibliometric and non-bibliometric RFs. For example, Table 4 shows the results we obtained with these machine learning algorithms for the applicants to the RF 11/E4 (level II).

Notice that we tested the performances of the learners only with respect to the not qualified class. We do that because we are mainly interested in understanding if we can use machine learning techniques to identify unsuccessful applicants who got not qualified. We are also reporting a limited amount of analysis data, specifically in this work we focus on precision and recall (and the related F-measure). Other aspects of the learners (such as the ROC curve) have been analyzed in our tests but they were always aligned with the results expressed by the three measures we are providing here. The results show that the best classifiers are those known to perform better on feature-rich datesets. In particular, SVM outperforms the other classification methods, and for this reason has been used in the rest of our analyses.

## Feature selection algorithm

In this section we describe the technique we used to analyze the relevance of the various predictors for classification purposes. The task consists in identifying a small set of predictors that allows to perform accurate predictions of the ASN results (RQ2). In case of a large number of predictors, several attribute engineering methods can be applied. The most widely adopted is attribute selection, whose objective is identifying a representative set of attributes from which to construct a classification model for a particular task. The reduction of the number of attributes can help learners that do not perform well with a large number of attributes. This also helps in reducing the computation time needed to create the predictive model.

There are two main classes of attribute selection algorithms: those who analyze the performance of the learner in the selection process (i.e., wrappers) and those who do not use the learner (i.e., filters). The first class is usually computationally expensive since the learner runs continuously to check how it performs when changing the attributes in the dataset. That leads to computation times that are two or more orders of magnitude larger compared to the learner itself. For this reason, we did only some limited experiments with learner-aware attribute selection. In our test cases, the results obtained were marginally better than those obtained with processes not using the learner. Consequently, we used a filter-based approach in our in-depth analysis.

We used Correlation-based Feature Selection (CFS) (*Hall & Holmes, 2003*), which is the first method that evaluates (and hence ranks) subsets of attributes rather than individual attributes. The central hypothesis of this approach is that good attribute sets contain attributes that are highly correlated with the class, yet uncorrelated with each other. At the heart of the algorithm is a subset evaluation heuristic that takes into account the usefulness of individual attributes for predicting the class along with the level of intercorrelation among them. The aforementioned technique has been used in the analysis presented in "Analysis of the Quantitative Indicators of Applicants".

## RESULTS

The aim of the analyses presented in this section is to answer the two Research Questions (RQs) discussed in "Introduction". Given the huge amount of data provided by the curricula of the applicants, we want to understand if machine learning techniques can be used to effectively distinguish between candidates who got the habilitation and those who did not (RQ1). We are also interested in identifying a small set of predictors that can be used to perform accurate predictions for the different RFs and scientific levels of the ASN (RQ2). We conclude this section with an assessment of the predictive power of our approach, in which we compare our best models with those that have been proposed in the literature to solve similar problems.

### Analysis of the recruitment fields and areas

The objective of the first experiment is to predict the results of the ASN (RQ1). We used SVM, which is the best machine learning algorithm emerged from the tests discussed in "Identification of the Prediction Algorithm". We classified our dataset with respect to the class of candidates who got the habilitation using the SVM learner. We first split the dataset into two partitions containing the data about candidates for level I and level II, respectively. For each partition, we classified separately the applicants of each RF. The results of our analysis are published in *Poggi et al. (2018a)*, and are summarized by the boxplots in Fig. 2. The boxplot is a method for graphically depicting the distribution of data through their quartiles. The central rectangle spans the first quartile to the third quartile. The segment inside the rectangle shows the median, and "whiskers" above and below the box show the locations of the minimum and maximum.

From these results, we observe that the performance of the learners for bibliometric and non-bibliometric RFs are very similar, and that they are distributed evenly (i.e., there is not

Performance of the SVM algorithm - 326 predictors

**Figure 2** **Boxplots depicting the performance of the SVM algorithm for academic level I and II.** Precision, Recall and F-measure values are reported for bibliometric (A, C) and non-bibliometric (B, D) RFs.

a polarization of bibliometric and non-bibliometric RFs). Moreover, we note that 154/184 (83.7%) and 162/184 (88%) RFs have F-measure scores higher than 0.6 for professional level I and II, respectively.

We also investigated the performance of the SVM learner on the data partitioned in the scientific areas in which RFs are organized. To do so, we split the dataset into 16 partitions:

nine for bibliometric SAs (01-09), one for the macro sector 11/E (Psicology) which is bibliometric, five for non-bibliometric SAs (10-14), and one for the RFs 08/C1, 08/D1, 08/E1, 08/E2 and 08/F1 which are non-bibliometric.

The results for both professional levels are summarized in Fig. 3, and the whole data are reported in *Poggi et al. (2018a)*. Also in this case, results are very accurate for both bibliometric and non-bibliometric disciplines, with F-measure scores spanning from a minimum of 0.622 (07-AVM) and 0.640 (02-PHY) for professionals level I and II, and a maximum of 0.820 (11-HPP) and 0.838 (14-PSS) for professional levels I and II. We observe that, at the associate professor level, the performance for non-bibliometric SAs (Fig. 3D) are significantly better than for bibliometric SAs (Fig. 3C). Moreover, the variance of the values is much lower for non-bibliometric SAs, as showed by the boxplots which are significantly more compressed.

### Analysis of the quantitative indicators of applicants

The objective of the next experiment is to identify a small set of predictors that allows to perform accurate predictions of the ASN results (RQ2). To this end, we analyzed the relevance of the various predictors for classification purposes using the CFS algorithm described in ''Feature Selection Algorithm''. The first step of our investigation consists of splitting our training set into partitions corresponding to the two professional levels of the ASN, and running the CFS filters on the data of each RF. We then produced a ranking of the selected predictors by counting the occurrences of each of them in the results of the previous computation. Figure 4 reports the top 15 predictors for the two professional levels considered.

We used the best overall learner emerged from the aforementioned tests (i.e., SVM) and applied it, for each academic level and RF, considering the top 15 predictors. The results of our analysis on the 184 RFs are summarized in Fig. 5, and the whole data are reported in *Poggi et al. (2018a)*. We observe that there has been a slight improvement in performances if compared to those obtained using all the predictors: 162/184 (88%) and 163/184 (88.6%) RFs have F-measure scores higher than 0.6 for professional level I and II, respectively. Moreover, also in this case, the results for bibliometric and non-bibliometric RFs are similar. An analysis of the indicators selected as top 15 predictors is presented in 'Discussion'.

### Evaluation

In order to assess the predictive power of our approach, in this section we compare our best models with those that have been proposed in the literature to solve similar problems. As discussed in ''Related Work'', three works are particularly relevant for this task: Vieira's model (*2014a*) based on rank ordered regression, Jensen's binomial regression model (*2009*), and the models developed by *Tregellas et al.* (*2018*).

A first analysis can be performed comparing the information summarized in Table 1 about the sizes of the datasets and the scopes of these works with our investigation. By considering the number of authors and papers, we observe that our dataset is some orders of magnitude greater than the others: i.e., 59,149 authors (our work) vs. 174 (Vieira), 3,659

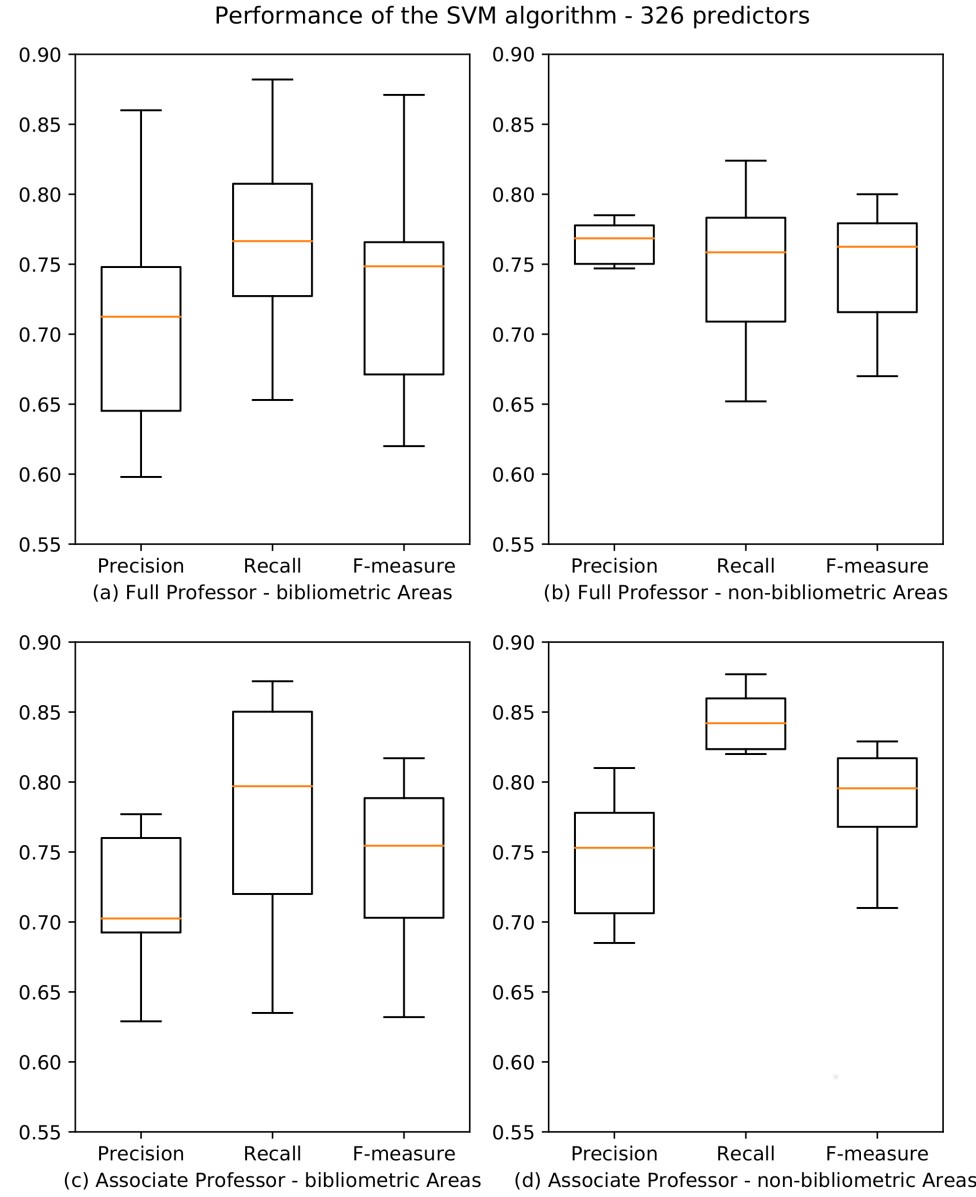

**Figure 3  Boxplots depicting the performance of the SVM algorithm for academic level I and II.** Precision, Recall and F-measure values are reported for bibliometric (A, C) and non-bibliometric (B, D) SAs.

(Jensen) and 363 (Tregellas) authors; 1,910,873 papers (our work) vs. 7,654 papers (Vieira). We also remark that Vieira's (_2014a_) and Tregellas's (_2018_) work are limited to very small samples of researchers from Portugal and the United States, while our and Jensen's works analyze a nationwide population. Moreover, while the other works focused on a limited set of indicators (Vieira's (_2014a_) model is based on three indicators, Jensen's (_2009_) on eight and Tregellas's (_2018_) on ten), we extracted a richer set of indicators from candidates' CVs (326 predictors). We also observe that, while our work and Jensen's (_2009_) cover all the disciplines, Vieira (_2014a_) limits the analysis to seven disciplines in hard sciences, and

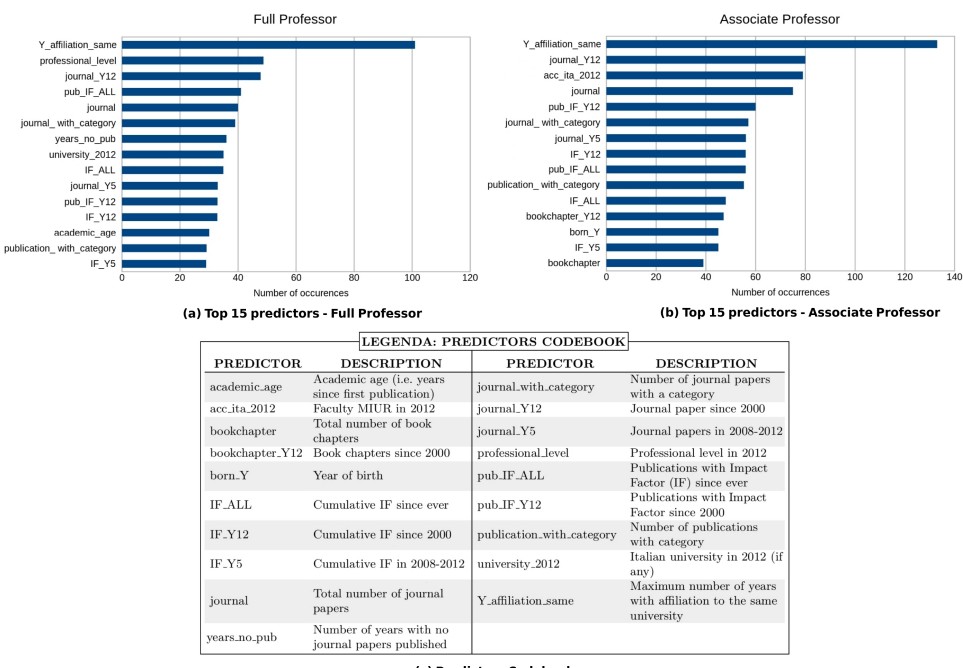

(a) Top 15 predictors - Full Professor

(b) Top 15 predictors - Associate Professor

| LEGENDA: PREDICTORS CODEBOOK | | | |
|---|---|---|---|
| **PREDICTOR** | **DESCRIPTION** | **PREDICTOR** | **DESCRIPTION** |
| academic_age | Academic age (i.e. years since first publication) | journal_with_category | Number of journal papers with a category |
| acc_ita_2012 | Faculty MIUR in 2012 | journal_Y12 | Journal paper since 2000 |
| bookchapter | Total number of book chapters | journal_Y5 | Journal papers in 2008-2012 |
| bookchapter_Y12 | Book chapters since 2000 | professional_level | Professional level in 2012 |
| born_Y | Year of birth | pub_IF_ALL | Publications with Impact Factor (IF) since ever |
| IF_ALL | Cumulative IF since ever | pub_IF_Y12 | Publications with Impact Factor since 2000 |
| IF_Y12 | Cumulative IF since 2000 | publication_with_category | Number of publications with category |
| IF_Y5 | Cumulative IF in 2008-2012 | university_2012 | Italian university in 2012 (if any) |
| journal | Total number of journal papers | Y_affiliation_same | Maximum number of years with affiliation to the same university |
| years_no_pub | Number of years with no journal papers published | | |

(c) Predictors Codebook

**Figure 4  Top 15 predictors selected by the CFS filter for professional level I (A) and II (B).** The $x$-axis shows how many times the predictors have been chosen by the CFS algorithm.

Tregellas (*2018*) to biomedical sciences. Overall, our dataset is very wide and rich, and less exposed to issues (e.g., biases) than those used in the other three works.

In order to evaluate the predictive power of our approach, we have to compare its performances with those of the aforementioned works. For this purpose, all the proposed predictive models must be tested on the same data. Since none of the datasets used in the considered works are freely available, we decided to test the models on representative samples extracted from our dataset, and compare the results with our approach.

The first model proposed by Vieira (*2014a*) is based on a composite predictor that encompasses 12 standard bibliometric indicators and that is obtained through factor analysis. Unfortunately, the authors don't provide a definition of such composite predictor, nor they discuss the details on how it has been computed. Given the lack of such information, we observed that is impossible to replicate the model and decided to exclude Vieira's (*2014a*) model from this experiment.

Jensen's (*2009*) model is a binomial regression model based on eight indicators: h, $h_y$, number of publications and citations, mean citations/paper, h/number of papers, age and gender. We decided to focus this analysis on the applicants to the associate professor level for two RFs: Informatics (01/B1) and Economics (13/A1). These two RFs have been chosen as representatives of bibliometric and non-bibliometric recruitments fields because they best meet two important criteria: (i) they received a very high number of applications; (ii) the two populations (i.e., those who attained the habilitation and those who did not attained it) are well balanced. For the same

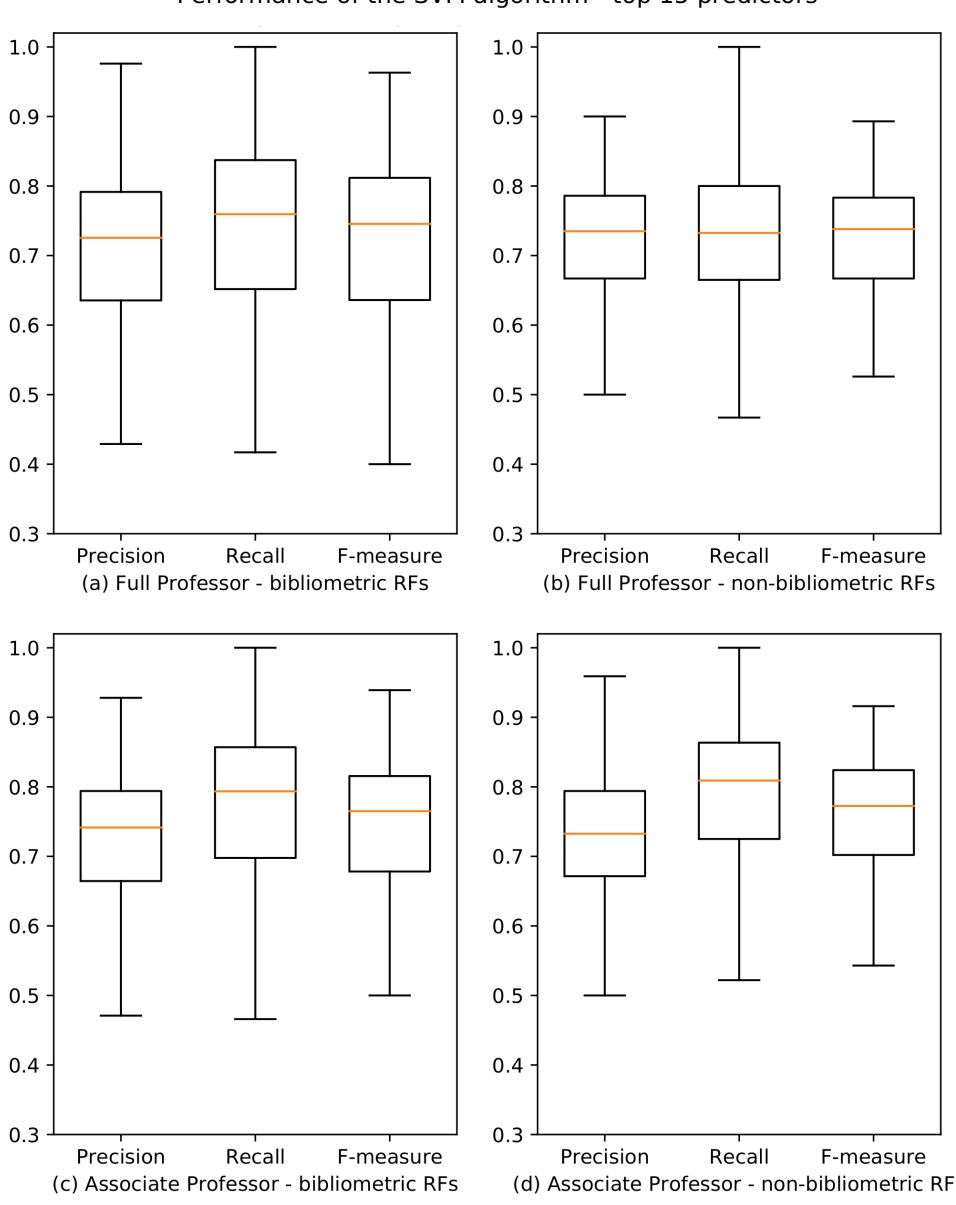

**Figure 5** **Boxplots depicting the performance of the SVM algorithm for academic level I and II using the top 15 predictors.** Precision, Recall and F-measure values are reported for bibliometric (A, C) and non-bibliometric (B, D) RFs.

reason we also considered the SAs "Mathematics and Computer Science" (MCS-01, bibliometric) and "Economics and Statistics" (ECS-13, non-bibliometric). In this way we are able to assess the predictive power of the models at different levels of granularity, both for bibliometric and non-bibliometric RFs and SAs. Since the indicators used by Jensen's (*2009*) models that were not present in our dataset (i.e., mean citations/paper, h/number of papers) could be derived from our data, we computed and added them to

**Table 5  Comparison of the performances of our models (OUR-SVM) with Jensen's (*2009*) models using eight predictors (J-LOG8) and one predictor (J-LOG1). Best Precision, Recall and F-measure values are in bold.**

| Field/ | Precision | | | Recall | | | F-measure | | |
|---|---|---|---|---|---|---|---|---|---|
| Area | J-LOG8 | J-LOG1 | OUR-SVM | J-LOG8 | J-LOG1 | OUR-SVM | J-LOG8 | J-LOG1 | OUR-SVM |
| 01/B1 | 0.592 | 0.611 | **0.718** | 0.588 | 0.578 | **0.773** | 0.590 | 0.594 | **0.744** |
| 13/A1 | 0.611 | 0.635 | **0.724** | 0.683 | 0.579 | **0.787** | 0.672 | 0.606 | **0.754** |
| MCS-01 | 0.677 | 0.638 | **0.692** | 0.719 | **0.782** | 0.753 | 0.697 | 0.703 | **0.721** |
| ECS-13 | 0.676 | 0.633 | **0.685** | 0.705 | 0.658 | **0.736** | 0.690 | 0.645 | **0.710** |

the test dataset. We then built the regression models using the aforementioned eight indicators and, as suggested by the authors, we also repeated the experiment using only the h-index, which has been identified as the one with the highest relevance. The results obtained by Jensen's (*2009*) models and our models are reported in Table 5.

The results show that our approach outperforms Jensen's regression models in all the considered RFs and SAs. The only exception is the recall value of the regression model based on the only h-index (LOG1) for the MCS-01 area. However, we report that the relative F-measure, which is a measure of the overall model accuracy, is much lower than our model. This can be explained by considering the low model precision, which is probably caused by an high number of false positives.

By comparing the F-measure values of the models we also observe that the regression models have the worst performances in non-bibliometric fields and areas (i.e., RF 13/A1 and SA ECS-13). The main reason is that the quantitative indicators used by the Jensen's (*2009*) models, which are mostly bibliometric, do not provide enough information for performing accurate predictions for non-bibliometric disciplines. In contrast, our approach is more stable, and leads to similar results in all RFs and SAs. The ability of our model to manage the variability of the different disciplines can be explained by the richness of the dataset on which the model is based.

We also compared the performance of our approach with *Tregellas et al.*'s (*2018*) two best models based on three indicators: sex, date of graduation, and number of first-author papers. As in the previous experiment, we decided to perform the test on two RFs, one bibliometric and one non-bibliometric, following the aforementioned criteria. As representative of bibliometric RFs we chose "Molecular biology" (05/E2) since *Tregellas et al.*'s (*2018*) work focused on the biomedical domain, and "Economics" (13/A1) as representative of non-bibliometric RFs (as in the previous experiment). Two out of the three indicators used by Tregella's models were not present in our dataset: number of first-author papers and date of graduation. While the first indicator can be easily computed using the publication list in the candidates' CVs, the latter (i.e., date of graduation) has to be gathered from external sources. Unfortunately, no freely-available database contains this information. We then had to search the web for authoritative sources (such as professional CVs, personal web pages, etc.) and manually process them to find information about the candidates' education. For this reason, we decided to focus our analysis on a sample of 50 randomly selected candidates for each of the considered RF. The output test dataset has

**Table 6  Comparison of the performances of our model (OUR-SVM) with *Tregellas et al.*'s (*2018*) two best models based on linear regression (T-LR) and support vector machines (T-SVM).** Best Precision, Recall and F-measure values are in bold.

| Field | Precision | | | Recall | | | F-measure | | |
|---|---|---|---|---|---|---|---|---|---|
| | T-LR | T-SVM | OUR-SVM | T-LR | T-SVM | OUR-SVM | T-LR | T-SVM | OUR-SVM |
| 05/E2 | 0.649 | 0.628 | **0.750** | 0.750 | **0.844** | 0.750 | 0.696 | 0.720 | **0.750** |
| 13/A1 | 0.440 | 0.550 | **0.690** | 0.393 | 0.393 | **0.645** | 0.415 | 0.458 | **0.667** |

been used for our experiment. The results of our model and *Tregellas et al.*'s (*2018*) models based on linear regression and SVM classifiers are reported in Table 6.

The results show that overall our approach outperforms Tregella's models. Also in this case there is an exception: the recall value of Tregella's model based on SVMs in RF 05/E2. However, by analyzing the relative F-measure, we note that Tregella's overall model accuracy is lower than our model: 0.720 for Tregella's SVM-based model, and 0.750 for our model. This is caused by the high number of false positives produced by Tregella's predictive model, which consequently results in lower precision and F-measure values compared to our model.

By comparing the F-measure values of the models we observe that Tregella's models have very low performances in the non-bibliometric RF (13/A1). We also note that, even considering the specific discipline for which Tregella's models have been designed for (i.e., RF 05/E2 - ''Molecular biology'', which is a discipline in the the biomedical domain), our model has better performances than two Tregella's regression models. This confirms that our approach is more stable and general, being able to perform accurate predictions in very different RFs and disciplines. As discussed in the previous experiment, the ability of our models to manage the variability and specificity of different disciplines can be explained by the richness of the features in our datasets, which have been automatically extracted from candidates' CVs, and that are fundamental to accurately predict the result of complex human processes (such as evaluation procedures).

## DISCUSSION

This research has been driven by the two research questions described in the introduction, and that can be summarized as follows:

- *RQ1:* Is it possible to predict the results of the ASN using only the information contained in the candidates' CVs?
- *RQ2:* Is it possible to identify a small set of predictors that can be used to predict the ASN results?

The analyses presented in 'Results' show that machine learning techniques can successfully resolve the binary classification problem of discerning between candidates that attained the habilitation and those who did not on the base of the huge amount of quantitative data extracted from applicants' CVs with a good accuracy. In fact, the results of the experiments for RQ1 have F-measure values higher 0.6 in 154/184 (83.7%) RFs and in 162/184 (88%) RFs for academic levels I and II, respectively. Moreover, the performances

are very similar and uniform for both bibliometric and non-bibliometric disciplines, and do not show a polarization of the results for the two classes of disciplines.

Through an attribute selection process we identified 15 top predictors, and the prediction models based on such predictors resulted to have F-measure values higher than 0.6 in 162/184 (88%) RFs and 163/184 (88.6%) RFs for academic levels I and II, respectively (RQ2). Also in this case, the results are uniform and equally distributed among bibliometric and non-bibliometric disciplines.

Some interesting considerations can be made by analyzing and comparing the top 15 predictors for the two academic levels (i.e., associate and full professor). First of all we remark that, as is obvious, many standard bibliometric indicators have been identified as relevant. In particular, seven of them are shared by both associate and full professor levels: the number of publications with impact factor since ever (`pub_IF_ALL`) and since 2000 (`pub_IF_Y12`), the number of publications with category (`publication_with_category`), the cumulative impact factor since ever (`IF_ALL`) and in 2008-12 (`IF_Y5`), and the number of journal papers since ever (`journal`) and since 2000 (`journal_Y12`) - see Fig. 4. However, we note that the first predictor (i.e., the one selected by the feature selection algorithm for most of the RFs) for both levels is `Y_affiliation_same` (i.e., the maximum number of years with affiliation to the same university). This is a non-bibliometric indicator which has not been considered by any of the papers reviewed in the 'Related Work'. We note that this result is coherent with the Italian model of academic careers, which is typically linear and inbreeding-based, meaning that most academics use to stay in the same university from basic studies up to the research career (*Aittola et al., 2009*). We plan to further investigate the correlation between working for the same institutions and the success to the ASN, and to analyze if there are differences among disciplines.

We also remark that there are interesting observations that concern each of the two levels and highlight peculiar aspects of each of them. For instance, we note that the year of birth (`born_Y`) is among the top 15 predictors for associate professors and not for full professor, suggesting that the age may be a relevant feature for the success at the beginning of an Italian scholar's career. This result is analogous to the one presented in *Tregellas et al. (2018)*, in which a similar indicator (i.e., the date of graduation) is used for predicting career outcomes of young researchers. Conversely, `years_no_pub` (i.e., the number of years in which no papers written by the candidate has been published) is a relevant predictor for full professor and not for associate professor. An explanation of this fact is that evaluation committees may have considered continuity in publications as a relevant factor in the evaluation of candidates to the full professor level (e.g., for discerning between candidates who have been active throughout their careers, and those who have not always been productive). Also, in this case, we plan to perform a deeper analysis of this point as future work.

An evaluation of the predictive power of our approach has been performed by comparing the results of our models with the best models that have been proposed in the literature to predict academic promotions. The comparison shows that our model outperforms Jensens' (*2009*) binomial regression models and Tregella's models on both bibliometric and non-bibliometric disciplines. This outcome proves that it is possible to predict with a good

accuracy the results of complex human processes such as peer-review assessments through computational methods. Moreover, the performance difference between the approaches is more evident for non-bibliometric disciplines. We observe that the outperformances of our results (overall and for non-bibliometric disciplines) are a straight consequence of the richness and quality of the predictors extracted form candidates' CVs. An explanation is that models which are mostly based on bibliometric indicators are not able to fully catch and explain all the different factors (e.g., cultural, social, contextual, scientific, etc.) that play a key role in peer-review evaluation processes.

## CONCLUSIONS

The results of this work are encouraging. We remark that the final goal of our work is not substituting evaluation committees by algorithms, but providing tools for supporting candidates, evaluators and policy makers involved in complex assessment processes such as the ASN. A candidate may use our system to self-evaluate his/her CV. Committee members could evaluate the consistency of their decisions across different evaluation sessions. In case of an appeal by rejected candidates to a higher panel, the panel itself could exploit our approach to analyze anomalies. Our system could also be useful for a foreign scholar who could get insight about how his CV is competitive against the Italian benchmarks. Also, policy makers could benefit from a system based on machine learning techniques such as the one presented in this paper in their decisions. At the local level, department heads and university presidents may evaluate people to recruit by guessing if they would be habilitated, since there are incentives. At the national level, the government may consider the results of our analysis to simplify the evaluation process. For instance, it could reduce the paperwork focusing on factors we identified as more relevant. Moreover, as already discussed, our approach would help committee members to minimize anomalies in their decisions. This would have the benefit of minimizing the number of requests of reviews and appeals, saving the time of both academic and administrative staff. Future directions of this research line consists in extending our analysis to more recent sessions of the ASN, and to analyze the impact of mobility on the career of academics. It would also be interesting to consider the applicants that have not been correctly classified by the learner in order to improve the approach and also have a more precise understanding of the factors that have been more relevant for assessments of academics performed by humans such as the ASN.

## ACKNOWLEDGEMENTS

We thank Andrea Bonaccorsi (University of Pisa) and Riccardo Fini (University of Bologna), who provided important considerations and discussions on this work. We would also thank the reviewers for their insightful comments.

### Funding

This research has been supported by the Italian National Agency for the Assessment of Universities and Research (ANVUR) within the Uniform Representation of Curricular Attributes (URCA) project (see articolo 4 of the 'Concorso Pubblico di Idee di Ricerca' - bando ANVUR, 12 February 2015). Paolo Ciancarini was also supported by CINI (ENAV project) and by CNR-ISTC. There was no additional external funding received for this study. The funders had no role in study design, data collection and analysis, decision to publish, or preparation of the manuscript.

### Grant Disclosures

The following grant information was disclosed by the authors:
Italian National Agency for the Assessment of Universities and Research (ANVUR).
CINI (ENAV project).
CNR-ISTC.

### Competing Interests

Silvio Peroni is an Academic Editor for PeerJ Computer Science.

### Author Contributions

- Francesco Poggi conceived and designed the experiments, performed the experiments, analyzed the data, contributed reagents/materials/analysis tools, prepared figures and/or tables, performed the computation work, authored or reviewed drafts of the paper, approved the final draft. Francesco Poggi is the main contributor of this work and the principal investigator of the project URCA that supported the research presented in this paper.
- Paolo Ciancarini and Silvio Peroni authored or reviewed drafts of the paper, approved the final draft.
- Aldo Gangemi analyzed the data, authored or reviewed drafts of the paper.
- Andrea Giovanni Nuzzolese and Valentina Presutti authored or reviewed drafts of the paper.

### Data Availability

The following GitHub repository contains raw data and code:

https://github.com/sosgang/asn2012-analysis.

The data/input/ folder contains the input data, and the data/output/ folder contains the output data of the analyses and experiments.

The src/ folder contains the Java source code used to perform the analyses and experiments discussed in the paper. The target folder contains a packaged runnable JAR to execute the analyses/experiments.

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
