# Peer review of "Predicting the results of evaluation procedures of academics"

_PeerJ Computer Science, doi:10.7717/peerj-cs.199_

## Round 0.1 · original submission · Minor Revisions

=========================

Reviewer 1 ·

Basic reporting

The paper “Predicting the results of evaluation procedures of academics” aims to assess whether it is possible i) to predict the results of the Italian Habilitation (ASN) using only the information contained in the candidates’ CVs?; ii) to identify a small set of predictors of the ASN results?
The paper is successful in reaching its targets and I admire the capacity of the author(s) of developing a very wide and rich dataset, which is less exposed to biases, as they stress in the comparative evaluation of their model.

Experimental design

Barring from being an expert of machine learning algorithms and techniques, the paper and its findings seem convincing.

Validity of the findings

I have just few minor tips for the authors.
First, discussion and conclusions should stress not only the prediction ability of the model developed, but also discuss the policy and practical implications of their findings for policy-makers. For instance, the maximum number of years with affiliation to the same university for obtaining ASN is a finding that contradicts the literature stream that stresses mobility as a positive feature in the academic career. Some kinds of tentative explanations of this unexpected results, maybe related to the specificities of the Italian context wherein the analysis took place, would be suggested.
Also practical implications of the findings should be stressed. E.g. Should the policy makers rely more machine learning techniques in order to make habilitation choice thus saving time of academic staff from the time-consuming peer review?
Finally, a very minor point, I would suggest in the Discussion and Conclusions section to replace the name of variables as defined in Figure 4, with the correspondent description to facilitate the reading.

·

Basic reporting

While the research presented here is of very high quality, the structure of the paper can however be improved on various accounts. For once, while the literature review is both impressive and thorough, I wonder if the "prediction of bibliometric indicators" subsection is absolutely necessary here, as the works discussed in detail there are of no "direct" relevance to the research and isn't referred to or used as benchmark in the "evaluation" section. I suggest that the authors sum up this subsection in a few lines, add it to the introductory paragraph of the "Related Work" section, and remove the "Prediction of the Results of Evaluation Procedures" subsection title. Also, the "results" section contain methodological procedures that do not contain any results. We suggest that the subsections contained in the "results" section be moved to the "methods and material" section, in which they clearly belong, and that the "evaluation" section be then renamed "results". Finally, while the content of the "Italian Scientific Habilitation" section is of crucial importance to the paper, I am not sure whether devoting a distinct section to it is the best option: some of its content clearly belongs to the introduction section, while the segment on bibliometric and non-bibliometric indicators would clearly fit in the "Methods and Material" section. In light of the above comments, I think that this paper would benefit from sticking closer to the IMRaD structural standard.

Experimental design

Additional information on why the 2012 ASN data is a representative and exhaustive sample of the whole population would have been welcome.

Validity of the findings

No comment.

Reviewer 3 ·

Basic reporting

In this manuscript, the authors develop a data mining approach to predict the results of ASN and identify some important features. The data used in this paper contains 59,149 researchers and 1,910,873 papers spanning in 184 recruitment field, which is relatively larger compared to related works. The treatment of the problem is technically sound.
1 - The acronym of Recruitment Field and Random Forests are both “RF”. I think this may lead to ambiguity.
2 – The quality of Figure 1 needs to be improved.

Experimental design

no comment

Validity of the findings

no comment

Additional comments

The subject and the proposed method are valuable. The problems and solutions in the paper are clearly stated. So, I believe that the article is suitable to be published in "PeerJ Computer Science journal" after a little revision.

---

## Round 0.2 · accepted · Accept

=====================================================

Reviewer 1 ·

Basic reporting

The review implements the changes requested in the first-round of the review process. So, I believe that the article is suitable to be published as it is.

Experimental design

The review implements the changes requested in the first-round of the review process.

Validity of the findings

The review explains better the policy and practical implications of their findings for policy-makers.

Reviewer 3 ·

Basic reporting

The paper has been revised well. All my concerns have been solved.

Experimental design

N/A

Validity of the findings

N/A

Additional comments

N/A

---

## Author Rebuttal · Round 0.2

**Department of Computer Science and Engineering**
University of Bologna
Italy

[Figure]

April 18th, 2019

Dear editor,
We thank both you and the reviewers for your work. We have reworked the paper following the reviewers' suggestions. In this letter we describe how we have modified the paper and addressed the comments we received. We report all comments; our answers are in bold.
We believe that the manuscript is now ready for publication.

Francesco Poggi

On behalf of all authors
* * *
Reviewer 1 (Anonymous)

Validity of the findings
I have just few minor tips for the authors.
First, discussion and conclusions should stress not only the prediction ability of the model developed, but also discuss the policy and practical implications of their findings for policy-makers.

> **1.** For instance, the maximum number of years with affiliation to the same university for obtaining ASN is a finding that contradicts the literature stream that stresses mobility as a positive feature in the academic career. Some kinds of tentative explanations of this unexpected results, maybe related to the specificities of the Italian context wherein the analysis took place, would be suggested.

**Agreed. We provided our explanation in Section "Discussion" (see lines 605-611 in our review PDF)**

> **2.** Also practical implications of the findings should be stressed. E.g. Should the policy makers rely more machine learning techniques in order to make habilitation choice thus saving time of academic staff from the time-consuming peer review?

**We added a discussion of the main practical implications of our findings for candidates, committee members and policy makers in Section "Conclusions" (see lines 635-636 in our review PDF)**

    **3.** Finally, a very minor point, I would suggest in the Discussion and Conclusions section to replace the name of variables as defined in Figure 4, with the correspondent description to facilitate the reading.

**Agreed. We renamed the variables in the Discussion section using their descriptions, as suggested (see lines 601-623 in our review PDF).**
* * *
Reviewer 2 (Maxime Sainte-Marie)

Basic reporting

    **4.** While the research presented here is of very high quality, the structure of the paper can however be improved on various accounts.

    For once, while the literature review is both impressive and thorough, I wonder if the "prediction of bibliometric indicators" subsection is absolutely necessary here, as the works discussed in detail there are of no "direct" relevance to the research and isn't referred to or used as benchmark in the "evaluation" section.

    I suggest that the authors sum up this subsection in a few lines, add it to the introductory paragraph of the "Related Work" section, and remove the "Prediction of the Results of Evaluation Procedures" subsection title.

**Agreed. We shortened the subsection "prediction of bibliometric indicators" and moved it as the introductory paragraph of the "Related Work" section, as suggested. We also removed the subsection titles (see lines 122-169 in our review PDF).**

    **5.** Also, the "results" section contain methodological procedures that do not contain any results. We suggest that the subsections contained in the "results" section be moved to the "methods and material" section, in which they clearly belong, and that the "evaluation" section be then renamed "results".

**Agreed. We moved the description of the techniques from the "Results" Section to the "Methods and Materials" Section (lines 400-440 and 468-486 in our review PDF have been moved to Subsection "Identification of the Prediction Algorithm" and "Feature Selection Algorithm" in the reviewed version). The updated version of "Results" contains the results answering Research Question #1 (see Subsection "Analysis of the Recruitment Fields and Areas") and Research Question #2 (see Subsection "Analysis of the Quantitative Indicators of Applicants"), and the evaluation of these results (see Subsection "Evaluation") as requested.**

    **6.** Finally, while the content of the "Italian Scientific Habilitation" section is of crucial importance to the paper, I am not sure whether devoting a distinct section to it is the best option: some of its content clearly belongs to the introduction section, while the segment on bibliometric and non-bibliometric indicators would clearly fit in the "Methods and Material" section.

**We agree with the reviewer that the content of the "Italian Scientific Habilitation" section is important, and that most of the information contained in this section fits the "Methods and Material" section. Providing all background details about the Italian Scientific Habilitation in a single place facilitates the reader in understanding the context of the case study. We renamed the section to "Data from the Italian Scientific Habilitation", and moved it to the beginning of "Methods and Materials" Section (see lines 234-300 in our review PDF)**

> **7.** In light of the above comments, I think that this paper would benefit from sticking closer to the IMRaD structural standard.

**Agreed. We restructured the paper as suggested into IMRaD structure, plus a "Related Work" section.**

> Experimental design
>
> **8.** Additional information on why the 2012 ASN data is a representative and exhaustive sample of the whole population would have been welcome.

**Agreed. We motivated the representativity of our sample data (see lines 310-311 in the review PDF).**

Validity of the findings
No comment.
* * *
Reviewer 3 (Anonymous)

Basic reporting
In this manuscript, the authors develop a data mining approach to predict the results of ASN and identify some important features. The data used in this paper contains 59,149 researchers and 1,910,873 papers spanning in 184 recruitment field, which is relatively larger compared to related works. The treatment of the problem is technically sound.

> **11 -** The acronym of Recruitment Field and Random Forests are both "RF". I think this may lead to ambiguity.

**We replaced the acronym "RF" used for Random Forest with "RandF" to avoid ambiguity (see Table 4 and line 422 in the review PDF).**

> **12 –** The quality of Figure 1 needs to be improved.

**We improved Figure 1 as follows:**
- **We broadened/stretched the rectangles and removed the abbreviations to improve readability**
- **We removed the light blue background in the circles to improve readability**
- **We exported the image in higher resolution (300 DPI)**

Experimental design
no comment

Validity of the findings
no comment

Comments for the Author
The subject and the proposed method are valuable. The problems and solutions in the paper are clearly stated. So, I believe that the article is suitable to be published in "PeerJ Computer Science journal" after a little revision.

Article ID: 33964